# Effects of Inulin Propionate Ester Incorporated into Palatable Food Products on Appetite and Resting Energy Expenditure: A Randomised Crossover Study

**DOI:** 10.3390/nu11040861

**Published:** 2019-04-16

**Authors:** Claire S Byrne, Edward S Chambers, Tom Preston, Catriona Tedford, Jerusa Brignardello, Isabel Garcia-Perez, Elaine Holmes, Gareth A Wallis, Douglas J Morrison, Gary S Frost

**Affiliations:** 1Section for Nutrition Research, Division of Diabetes, Endocrinology and Metabolism, Faculty of Medicine, Imperial College London, Hammersmith Hospital, London W12 0NN, UK; claire.byrne@imperial.ac.uk (C.S.B.); e.chambers@imperial.ac.uk (E.S.C.); 2Stable Isotope Biochemistry Laboratory, Scottish Universities Environmental Research Centre, University of Glasgow, East Kilbride, Glasgow G75 0QF, Scotland; Tom.Preston@glasgow.ac.uk (T.P.); Douglas.Morrison@glasgow.ac.uk (D.J.M.); 3School of Computing, Engineering and Physical Sciences, University of the West of Scotland, Paisley Campus, Paisley PA1 2BE, Scotland; Catriona.Tedford@uws.ac.uk; 4Department of Surgery and Cancer, Computational and Systems Medicine, Imperial College London, South Kensington Campus, London SW7 2AZ, UK; j.brignardello-guerra15@imperial.ac.uk (J.B.); i.garcia-perez@imperial.ac.uk (I.G.-P.); elaine.holmes@imperial.ac.uk (E.H.); 5Institute of Health Futures, Murdoch University, South Street, Western Australia 6150, Australia; 6School of Sport, Exercise and Rehabilitation Sciences, University of Birmingham, Edgbaston, Birmingham B15 2TT, UK; g.a.wallis@bham.ac.uk

**Keywords:** SCFA, propionate, appetite, energy intake, energy expenditure

## Abstract

Supplementation with inulin-propionate ester (IPE), which delivers propionate to the colon, suppresses ad libitum energy intake and stimulates the release of satiety hormones acutely in humans, and prevents weight gain. In order to determine whether IPE remains effective when incorporated into food products (FP), IPE needs to be added to a widely accepted food system. A bread roll and fruit smoothie were produced. Twenty-one healthy overweight and obese humans participated. Participants attended an acclimatisation visit and a control visit where they consumed un-supplemented food products (FP). Participants then consumed supplemented-FP, containing 10 g/d inulin or IPE for six days followed by a post-supplementation visit in a randomised crossover design. On study visits, supplemented-FP were consumed for the seventh time and ad libitum energy intake was assessed 420 min later. Blood samples were collected to assess hormones and metabolites. Resting energy expenditure (REE) was measured using indirect calorimetry. Taste and appearance ratings were similar between FP. Ad libitum energy intake was significantly different between treatments, due to a decreased intake following IPE-FP. These observations were not related to changes in blood hormones and metabolites. There was an increase in REE following IPE-FP. However, this effect was lost after correcting for changes in fat free mass. Our results suggest that IPE suppresses appetite and may alter REE following its incorporation into palatable food products.

## 1. Introduction

A number of epidemiological studies have shown that increased dietary fibre intake protects against weight gain [1,2,3]. Intervention studies in animals also support this observation [4,5]. It has been suggested that the positive effects associated with fibre are partly attributable to short-chain fatty acids (SCFA), products of fibre fermentation. The SCFA propionate has been shown to stimulate the release of the anorexigenic hormones peptide YY (PYY) and glucagon-like peptide-1 (GLP-1) from intestinal L-cells [6].

However, the degree of fermentation of dietary fibre and rate, site and extent of SCFA produced varies across individuals [7]. Moreover, the levels of dietary fibre intake needed to stimulate the release of gut hormones are not achievable in most Western societies [8]. Oral SCFA supplementation is not a feasible dietary intervention strategy in humans because the very poor organoleptic properties of SCFA prohibit consumption of the requisite doses for any significant duration. As a result, the majority of current available evidence demonstrating positive metabolic effects of raising gut-derived SCFAs has been obtained from animal models with few human studies. To overcome this problem we have developed a method of delivering SCFA to the colon, the site of maximal production in the GI tract [9], using inulin-SCFA esters. We have previously shown that selectively increasing propionate delivery to the colon using inulin-propionate ester (IPE) reduces energy intake and increases circulating PYY and GLP-1 concentrations acutely in humans, and prevents weight gain [10].

There is also evidence to suggest that propionate supplementation can prevent weight gain independently of altered energy intake [10,11,12]. Oral propionate supplementation has been reported to protect against diet-induced obesity by enhancing energy expenditure (EE) via an increase in lipid oxidation in mice [11]. We have recently shown that acute oral sodium propionate supplementation also increases resting EE (REE) and lipid oxidation in humans [13]. However, the potential effect of IPE supplementation on energy expenditure and substrate oxidation has not previously been investigated.

In our previous studies, IPE was provided as a powder supplement which volunteers were asked to mix with their habitual diet. The incorporation of IPE into food products (FP) would not only standardise the way participants consume IPE for long-term studies but would also be an attractive way to deliver IPE on a population scale. An additional important question is whether IPE changes the palatability of common FP, since release of propionate in product preparation may “spoil” the product because of the poor organoleptic properties of propionate. Here, a bread roll and fruit smoothie were the FP used for supplementation of IPE. The aim of the present study was two-fold: (1) to confirm the appetite-suppressing effects of IPE in healthy overweight and obese humans when incorporated into palatable FP and; (2) to explore the effect of elevated colonic propionate on REE and substrate oxidation. We hypothesised that an increase in colonic propionate delivery following a seven-day supplementation period with IPE-FP would reduce ad libitum energy intake and this would be mediated by an increase in PYY and GLP-1.

## 2. Materials and Methods

This was a randomised, double-blind, crossover feeding study, which was approved by the London-Brent Research Ethics Committee (REC reference number 14/LO/0645) and registered on the ISRCTN registry (ISRCTN71814178). This study was conducted in accordance with the Declaration of Helsinki of 1975 and later revisions.

### 2.1. Participant Recruitment

Participants were recruited via existing volunteer databases and poster advertisements. Men and women aged between 18–65 years with body mass index (BMI; kg/m^2^) 25–40 were eligible for inclusion. Exclusion criteria included weight change of ≥3 kg in the preceding two months, current smokers, excess alcohol intake, substance abuse, any chronic illness or GI disorder, pregnancy and use of medications likely to interfere with energy metabolism, appetite regulation and hormonal balance, including: anti-inflammatory drugs or steroids, antibiotics, androgens, phenytoin, or thyroid hormones. Eligible participants were randomised into the study by the sealed envelope randomisation service (www.sealedenvelope.com).

### 2.2. Dietary Intervention

IPE, designed for targeted delivery of propionate to the colon, was the interventional supplement used in this study and was produced in-house as previously described [10,14]. Inulin (Beneo HP, DKSH, London, UK) was used as a positive control (10 g/d). Dietary supplements were incorporated into FP by Leatherhead Food Research (Epsom, Surrey, UK) in order to standardise how participants consumed them. A bread roll and fruit smoothie were the two FP chosen as the ‘vehicle’ for supplementation. Supplemented-FP each contained either 5 g of IPE or 5 g of inulin, thus providing a total dose of 10 g/d when consumed together. Unsupplemented FP (containing neither IPE nor inulin) were also produced for Control visits. Apart for the addition of dietary supplements to the FP, all FP were identical and were made in the same batch. The nutritional composition of food products, as assessed using DietPlan 6 (Forestfield Software Ltd., Horsham, West Sussex, UK), is shown in Appendix A. The FP were frozen post-production and were defrosted on the morning of consumption.

### 2.3. Study Design

An overview of the study design is outlined in Appendix A. This study involved 4 separate study visits: (1) an “acclimatization” study visit, (2) a “control” study visit, (3) a “post-supplementation study visit 1” following the first six-day feeding period, and (4) a final “post-supplementation study visit 2” following the second six-day feeding period. Participants received un-supplemented FP (control-FP) on the acclimatisation and control study visits with a seven-day washout period after both study visits. In a randomised order, participants then consumed FP supplemented with either IPE (IPE-FP) or inulin (inulin-FP) for 6 days followed by post-supplementation study visit 1, during which they received the supplemented-FP for the seventh and final time. Following a 14-day washout period, participants entered into the final six-day feeding period followed by post-supplementation study visit 2 where they consumed the FP that they had not yet received.

### 2.4. Supplementation Period

Per supplementation period, participants were provided with six bread rolls and six smoothies and asked to consume one of each product per day for the six days leading up to the post-supplementation study visit. Participants were provided with a supplementation diary in order track the time and date of product consumption and to monitor compliance.

### 2.5. Study Visit Preparation

Participants were asked to avoid caffeine, alcohol and strenuous exercise, and to record their food intake for the 24 h before each study visit. A standard meal of the participant’s choice was consumed the evening before each visit in order to promote consistency between visits. Participants were asked to arrive having fasted for 12 h prior to each study visit. Participants were also asked not to start any new diet or intensive exercise regimes during the study period in order to prevent conflicting results.

### 2.6. Study Visit Protocol

The study visit protocol is outlined in Appendix A. Study visits were conducted between October 2014 – February 2016 at the National Institute for Health Research (NIHR) Imperial Clinical Research Facility (CRF), Hammersmith Hospital, London, United Kingdom.

Prior to the consumption of the test meal (0 min), body composition measurements were assessed by bioelectrical impedance analysis (Tanita BC-418 analyser; Tanita Corporation, Tokyo, Japan). Participants were asked to empty their bladders and change into a set of hospital scrubs prior to the collection of body composition data, in order to promote consistency between visits. Participants completed an international physical activity questionnaire (IPAQ), which captured data related to physical activity levels during the previous seven days. Side effects experienced during the previous seven days were also assessed using a series of 100 mm visual analogue scales (VAS). The left extremity of the VAS was labelled with ‘not at all’ and the right-hand extremity was labelled with “severe problem”. In addition, a number of baseline samples were collected in duplicate (>5 min apart). Baseline blood samples were collected via a peripheral cannula in order to assess metabolite and hormone concentrations. Baseline breath H_2_ measurements were collected in real-time using a handheld breath H_2_ m (Gastro+ Gastrolyser Breath Hydrogen Monitor; Bedfont Scientific, Maidstone, Kent, UK). Baseline subjective feelings of appetite (“how hungry/full do you feel?”) and mood (“how nauseous do you feel?”) were also collected using a series of 100 mm VAS. The left extremity of the VAS was labelled with ‘not at all’ and the right-hand extremity was labelled with “extremely”. Participants were asked to draw a vertical line on the VAS depending on how intensely they were experiencing each feeling.

Overnight fasted REE and substrate oxidation was measured using open-circuit indirect calorimetry (Gas Exchange Monitor, GEM Nutrition, Daresbury, Cheshire, UK). During the collection of REE data, participants were asked to lie semi-supine and a transparent canopy was placed over their head and thorax. Participants were permitted to watch television, read or listen to music while under the canopy but were instructed to stay as still as possible. For the measurement of urinary nitrogen excretion, participants were asked to collect all urine in a collection container for 420 min after consuming the test meal.

Participants received a standardised breakfast (600 kcal; 114.6 g carbohydrate, 13.8 g fat, 15.6 g protein, 7.1 g fibre) at 0 min, which consisted of the appropriate bread roll and smoothie according to the participant’s randomisation along with breakfast cereal, milk and margarine. Following breakfast, participants completed a product satisfaction questionnaire where they rated the product overall as well as based on its appearance, aroma, flavour and texture. Participants ticked a box between 1 and 9, where 1 was labelled with “dislike extremely”, 5 was labelled with “neither like nor dislike” and 9 was labelled with “like extremely”. REE was measured before (~160–175 min) and after (~210–240 min) a standardised snack (508 kcal, 57.1 g carbohydrate, 21.3 g fat, 20.7 g protein, 2.0 g fibre), which was served at 180 min.

Further venous blood samples (15, 30, 60, 90, 120, 180, 195, 210, 240, 270, 300, 360, 420 min), VAS (15, 30, 60, 90, 120, 180, 195, 240, 270, 300, 360, 420 min) and breath H_2_ concentration (60, 120, 180, 240, 300, 360 min) measurements were collected throughout the study visit.

After 420 min, the cannula was removed and participants were provided with an ad libitum meal. This time point was chosen based on previous data suggesting successful delivery of IPE to the colon at this time [10]. This was a savoury meal and participants were offered the choice of tomato and herb pasta (per 100 g: 129 kcal; 21.0 g carbohydrate, 3.2 g fat, 3.3 g protein, 1.8 g fibre) or four cheese pasta (per 100 g: 165 kcal; 21.5 g carbohydrate, 6.6 g fat, 4.3 g protein, 1.1 g fibre). Participants consumed the same choice of buffet meal on the control and post-supplementation study visits. Participants were instructed to eat until they felt comfortably full and the amount of food consumed was weighed. Following the ad libitum meal, participants were discharged from the CRF.

An acclimatisation study visit was added to the protocol in order to introduce, or acclimatise, participants to the study protocol [15]. The only difference between the acclimatisation study visit and the other study visits was that a small amount of blood (1 mL) was taken at each time point and was immediately discarded.

### 2.7. Blood Sample Preparation

Ten millilitres of blood was collected at each time point for assay of plasma glucose (BD Fluoride EDTA Vacutainer; BD, New Jersey, USA), serum insulin and NEFA (BD Serum SST Vacutainer; BD, New Jersey, USA) and plasma gut hormones (BD Lithium Heparin Vacutainer; BD, New Jersey, USA) containing 20 µL/mL whole blood Aprotinin pancreatic protease inhibitor, Nordic Pharma UK Ltd., Reading, UK) measurements. Plasma tubes were centrifuged immediately at 2500 RCF for 10 min at 4 °C. Serum tubes were allowed to clot before centrifugation. Resulting plasma/serum was aliquoted into Eppendorfs and frozen at −20 °C until analysis.

### 2.8. Ad Libitum Energy Intake

Food intake data were collected for 19 participants. One participant’s food was disposed of before obtaining data on food intake and one participant refused to consume any food at the buffet meals. Food intake (g) was multiplied by the energy density of the meal in order to calculate the mean energy intake at the ad libitum meal.

### 2.9. Visual Analogue Scales

A composite appetite score (CAS) was calculated using the following formula [16]: [Hunger + (100 − Fullness) + Desire to Eat + Appetite for Meal]/4.

### 2.10. Energy Expenditure and Substrate Oxidation Measurement

REE, RER, and substrate oxidation were estimated from oxygen consumption and carbon dioxide production using the following equations [17]:RER = V^·^CO_2_/V^·^O_2_
REE = [(3.91 ∗ V^·^O_2_) + (1.1 ∗ V^·^CO_2_) − (1.93 ∗ N)]
Carbohydrate oxidation = [(4.57 ∗ V^·^CO_2_) − (3.23 ∗ V^·^O_2_) − (2.6 ∗ N)
Fat oxidation = [(1.69 ∗ V^·^O_2_) − (1.69 ∗ V^·^CO_2_) − (2.03 ∗ N)]
N = [((((CH_4_N_2_O/16.6) ∗ 0.466) ∗ vol)/time) ∗ 1.2]
Protein oxidation = [N ∗ 6.25]
where RER is the respiratory exchange ratio, REE is in kcal/min, V^·^O_2_ is the rate of oxygen consumed in L/min, V^·^CO_2_ is the rate of carbon dioxide in L/min, N is urinary nitrogen in g/min, carbohydrate, fat and protein oxidation are in g/min, CH_4_N_2_O is urinary urea in mmol/L, vol is the urine volume in L and time is in min.

### 2.11. Metabolic and Hormone Analysis

A fasting blood sample was sent to the Dept. of Biochemistry, Hammersmith Hospital on each study visit for cholesterol (total, HDL, LDL) and triglyceride measurement. An aliquot of urine was also sent to the Dept. of Biochemistry for urinary urea quantification. Glucose analysis was performed in the Dept. of Biochemistry, Hammersmith Hospital using a ci8200 analyser enzymatic method (Abbott, Abbott Park, IL, USA). Insulin analysis was performed in-house using a human insulin RIA kit (Millipore Corporation, Billerica, MA, USA) according to manufacturer’s guidelines with 50 µL serum. PYY and GLP-1 were measured using previously established in-house specific and sensitive RIA [18,19]. Acetate, propionate and butyrate were measured at Dept. of Cancer and Surgery using an Agilent 7000C Triple Quadrupole GC/MS System (Agilent, Santa Clara, CA, USA) according to a previously published method [20]. NEFA were measured using a commercial kit (FA115 NEFA kit; Randox, London, UK) and were measured using an ILAB 650 Clinical Chemistry Analyser (Instrumentation Laboratory, Birchwood, Warrington, UK).

### 2.12. Statistics

It was estimated, using the G*Power power calculator, that 19 volunteers would be needed, based on a power of 80%, α = 0.05, to detect a difference in mean plasma PYY concentration (primary outcome measure) of 16 pmol/L with a standard deviation of 23 between treatments [10].

Data are presented as median [IQR]. The average of two separate measurements for breath H_2_, CAS, blood hormone and metabolite data, which were collected before the test meal, were used for the 0 min measurement, which is referred to as a baseline or fasting measurement. AUC were calculated using the trapezoidal rule and divided by the relevant time period (180 or 240 min).

Owing to the non-parametric nature of the majority of outcome measures and in order to determine the simple rank order of the three treatments (Control, Inulin and IPE), data were compared using a Friedman test with post-hoc Wilcoxon signed-rank tests. Significance was considered *p* < 0.05. Statistical analyses were performed using IBM SPSS Statistics v23-24 (Armonk, New York, USA). Graphs were prepared using GraphPad Prism v5.0-7.0 (San Diego, CA, USA).

## 3. Results

Twenty-three participants were randomised into this study with 21 participants completing all four study visits. Two participants withdrew from the study before attending their first study visit. Twelve participants were randomly assigned to receive IPE-FP during the first supplementation period and nine participants received inulin-FP first.

### 3.1. Participant Characteristics

An overview of participant characteristics on each study visit is given in Table 1. There were no differences in fasting blood lipid profiles, power of food scale, IPAQ values or in food intake in the 24 h prior to study visits. There were significant differences in body weight (*p* = 0.023) and FFM (*p* = 0.024) among treatments (Table 1). Post hoc tests revealed that body weight was significantly higher following IPE supplementation versus both inulin supplementation (*p* = 0.017) and control (*p* = 0.04). There was no significant difference in body weight (*p* = 1.0) between inulin supplementation and control. FFM was significantly higher following IPE supplementation (*p* = 0.001), when compared to control. Compared to inulin supplementation, there was a trend for FFM to be higher on the control visit (*p* = 0.073) and following IPE supplementation (*p* = 0.063). There was a significant difference in total body water (TBW; *p* = 0.026) between the three treatments (Table 1). Post hoc tests revealed that TBW was significantly higher following IPE supplementation versus both inulin supplementation (*p* = 0.045) and control (*p* = 0.003). There was no significant difference in body weight between inulin supplementation and control (*p* = 0.21).

### 3.2. Ad Libitum Energy Intake, Gut Hormone Concentrations and Composite Appetite Score

There was a significant difference in ad libitum energy intake between treatments (*p* = 0.012; Table 2). Post-hoc tests revealed that there was a trend for IPE to reduce food intake compared to control (*p* = 0.056) and inulin supplementation (*p* = 0.059) but there was no difference between control and inulin supplementation (*p* = 0.75). However, there were no differences in the baseline measurements or the AUC_0-420_ for peripheral PYY or GLP-1 concentrations or CAS between treatments (Table 2; Appendix A).

### 3.3. Food Product Taste and Appearance Ratings and Side Effects

There were no differences in bread roll or fruit smoothie taste and appearance ratings between treatments (*p* = 0.25–0.89) (Table 3). There was a significant difference in stomach discomfort, nausea, flatulence, heartburn and toilet frequency between treatments (Figure 1). Post hoc tests revealed that, compared to control, stomach discomfort, nausea, flatulence, heartburn and toilet frequency were all significantly increased following inulin (*p* = 0.001–0.021) and IPE (*p* = 0.004–0.043) supplementation. However, there was no significant difference in these side effects between supplements (*p* = 0.245–0.86). There was also a trend for a difference in stomach discomfort between treatments but this did not reach statistical significance (*p* = 0.053). There was no difference in bloating and belching between treatments.

### 3.4. Resting Energy Expenditure and Substrate Oxidation

There was a trend for a difference in fasting REE between treatments (Table 4). There was a significant difference in the AUC_0-240_ for postprandial REE between treatments. This was due to greater REE following IPE-FP when compared to inulin (*p* = 0.021) and control (*p* = 0.011). However, due to significant differences in body composition noted between treatments, REE corrected for FFM was calculated. After correcting for changes in FFM, there was no significant difference in REE between treatments (*p* = 0.22–0.37). There were no significant differences in RER or carbohydrate, lipid and protein oxidation between treatments (*p* = 0.33–0.95).

### 3.5. Breath H_2_

There was no difference in fasting breath H_2_ concentrations between treatments (*p* = 0.44; Table 5). There was a significant difference in the AUC_0-420_ and the AUC_24o-420_ for breath H_2_ between treatments. This was due to greater breath H_2_ following inulin supplementation versus control (AUC_0-420_
*p* = 0.002, AUC_24o-420_
*p* = 0.002) and IPE (AUC_0-420_
*p* = 0.027, AUC_24o-420_
*p* = 0.014) owing to the greater amount of fermentable carbohydrate in inulin-FP than in IPE-FP (10 g compared with 7.3 g) and control-FP. However, breath H_2_ concentrations were significantly elevated above baseline concentrations between 240 and 420 min following consumption of IPE-FP (*p* < 0.01; Appendix A). This suggests fermentation of IPE and release of propionate in the colon occurred in a time course similar to that previously reported [10,21]. Breath H_2_ concentrations were also consistently elevated above baseline concentrations between 240 and 420 min after receiving inulin-FP (*p* < 0.01).

### 3.6. Blood Hormones and Metabolites

There were no differences in the baseline measurements or the AUC_0-420_ for peripheral glucose, insulin, NEFA or SCFA concentrations between treatments (Table 5; Appendix A).

## 4. Discussion

Investigating the physiological impact of SCFA in humans has long been hampered by our inability to undertake meaningful intervention studies where we are able to control the site, rate and extent of SCFA production in the GI tract. Oral SCFA are unpalatable and are not sustainable as a dietary intervention strategy. Encapsulation or other routes of duodenal delivery are possible but whether small intestinal and large intestinal SCFA behave in a physiologically identical manner remains to be elucidated. However, as L cells that co-express GLP-1 and PYY increase in density along the GI tract it has been suggested that increasing colonic propionate delivery or production is more likely to lead to a suppression in appetite than increasing small intestinal propionate content [22]. Direct and controlled delivery of individual SCFA to the proximal colon, the site of maximal production in the GI tract, has been achieved with IPE and represents a step-change in our ability to investigate the role of SCFA in human health. In the present study, we incorporated a 10 g/d dose of IPE into FP in order to assess their effect on appetite and energy expenditure. IPE-supplemented-FP (bread roll + fruit smoothie) provided a 10 g/d dose of IPE when consumed together, which is the dose that has previously been shown to modulate appetite responses [10,14,21], and increase daily propionate production by 2.5-fold based on the average UK non-starch polysaccharide intake [10].

Previous studies have shown a reduction in ad libitum energy intake following the consumption of IPE [10,14,21]. As hypothesised, there was a significant difference in energy intake between treatments, which was driven by a greater suppression in appetite following IPE-FP versus both inulin-FP and control-FP (*p* = 0.056–0.059). Despite the difference in ad libitum energy intake, there was no difference in the CAS. However, this is in line with previous studies [10,21], suggesting that IPE does not suppress subjective appetite responses, but reduces energy intake, consistent with the action of a physiological satiation signal. It was originally hypothesised that any reduction in energy intake following IPE-FP would be mediated by an increase in PYY and GLP-1 concentrations, as elevated colonic propionate has previously been shown to stimulate gut hormone secretion acutely in humans [10]. However, we found no difference in GLP-1 or PYY concentrations following a seven-day IPE supplementation, which is in line with previous findings following long-term supplementation with IPE [10]. Other studies have also reported that non-digestible carbohydrates (NDC) consumption and increased colonic SCFA production improve body composition and reduce energy intake independent of changes in systemic gut hormone concentrations in rodents [23,24,25]. It is possible that we were not able to detect differences in gut hormone release as it has previously been highlighted that venous sampling measurements are not always reflective of gut hormone release into the portal vein [26]. However, alternative mechanisms may also be responsible. It is possible that these appetite suppressing effects may be a result of elevated portal propionate concentrations; portal propionate has been shown to induce vagal signalling via FFAR3 [12,27,28] and ruminant studies have shown an association between elevated portal propionate concentrations and a reduction in energy intake [29,30].

The results of the food product ratings demonstrate that IPE-FP were palatable to the participants. Furthermore, the gastrointestinal side effects noted following IPE-FP are comparable to those reported following supplementation with inulin-FP, and are common side effects associated with fibre consumption. Thus, the incorporation of IPE into FP allows for the delivery of IPE in conditions that more accurately reflect the way humans eat but also provides an attractive way to deliver IPE on a population scale.

Oral SCFA have previously been shown to protect against weight gain in HFD-fed mice without changing food intake or physical activity levels, which suggests that SCFA may affect EE and/or substrate oxidation [11,31,32]. Rectal propionate infusions and oral propionate supplementation have previously been reported to increase REE in humans [13,33]. In the present study there was a trend for a difference in fasted REE between treatments and a significant increase in postprandial REE following IPE-FP compared to inulin (0.103 kcal/min) and control (0.111 kcal/min). Theoretically, if this effect was sustained over a period of 24 h, this would result in an increase in EE of ~148–160 kcal/d. If consumed daily, this small increase in EE may be beneficial in helping prevent “weight creep”, which causes the average adult to gain 0.4–0.9 kg/year [34,35]. However, due to the differences in body composition between treatments, the REE was corrected for FFM, the primary determinant of REE [36]. After correcting for FFM, the effect was no longer significant. Colonic and rectal infusions of SCFA have been reported to increase lipid oxidation in humans [33,37]. In addition, oral supplementation with sodium propionate has been shown to increase lipid oxidation in humans [13]. It was, therefore, originally hypothesised that an increase in REE would be driven by a shift in substrate utilization towards fat oxidation. However, no significant difference in substrate oxidation was noted in the present study between treatments based on the indirect calorimetry data. As REE measurements were not collected for the whole 420 min postprandial sampling period, it is possible that significant differences in REE or substrate oxidation between supplementation treatments were missed. In addition, it is currently unknown whether increased SCFA delivery to the small versus large intestine have different physiological effects and warrants further investigation.

There was a significant difference in the AUC_0-420_ and the AUC_240–420_ for breath H_2_ between treatments, which was driven by higher breath H_2_ concentrations following the consumption of inulin-FP versus IPE-FP and control-FP [10]. However, this is a consistent observation for inulin compared with other fermentable carbohydrates and may also reflect that the 10g/day intended inulin dose delivered in inulin-FP contained more inulin than IPE-FP (~7g/day allowing for propionate content in 10 g IPE) and control-FP. Despite the differences between treatments, breath H_2_ concentrations were significantly elevated above baseline concentrations from 240–420 min to the end of the study visit following the consumption of IPE-FP and inulin-FP, suggesting fermentation of inulin and IPE share a similar time course and similar to that previously reported and at the time of the ad libitum meal [10].

There were no significant differences in serum SCFA concentrations between treatments, thus despite the delivery of a large amount of propionate to the colon this is not translated into significantly higher circulating propionate concentrations. However, this is unsurprising as propionate has been shown to be effectively sequestrated by the liver [38] and a recent report has highlighted that propionate levels in peripheral blood are an inaccurate reflection of the amounts being produced and absorbed from the gut [39]. Nevertheless, previously we have reported a significant increase in propionate ^13^C enrichment in the peripheral circulation following the consumption of a ^13^C labelled IPE variant, confirming that the bound propionate from IPE is absorbed from the gut and is available systemically [10].

As propionate is a potential gluconeogenic precursor [40], fasting and postprandial blood glucose and insulin concentrations were measured. There was no differences in the baseline values or the AUC for glucose, insulin or NEFA between treatments. However, these results are in line with previous studies [10,21].

Interestingly, body weight was significantly higher following IPE-FP versus control and inulin-FP. There were no differences between treatments in FM, however, FFM was significantly higher following IPE-FP versus control. There is some evidence that NDC supplementation can increase FFM [41] or preserve FFM during weight loss [42,43] in humans. However, due to the short seven day supplementation period it is unlikely that the differences observed is due to the IPE-FP. Furthermore, TBW was significantly higher following supplementation with IPE versus control and inulin-FP, which suggests that greater water retention following IPE may be driving the differences in body weight and composition noted. As there was a >14-day washout period between supplementation periods and body composition was not recorded prior to IPE supplementation, we cannot be certain that the observed effects on body weight or body composition were due to the relatively short supplementation period and thus these data must be treated with caution.

The addition of IPE to whole foods was a major strength to this study, providing opportunities for long-term intervention studies with IPE, which can be consumed as part of a normal habitual diet that is both palatable and versatile. However, further work is needed to understand whether IPE has the potential to increase REE. As propionate has previously been shown to stimulate sympathetic nervous system (SNS) activity resulting in an increase in EE via FFAR3 [44] it would be of interest to measure markers of SNS activity, such as heart rate, in order to investigate this as a possible mechanism for increased EE in future studies. A possible limitation in the current study is that participants were not counselled or advised on how to replace habitual dietary intake with the study foods. However, if the addition of FP resulted in the consumption of a surplus of calories during the supplementation periods, any effect of this is likely to be controlled for due to the crossover design.

## 5. Conclusions

IPE added to common food products does not influence palatability. The increased colonic propionate delivery reduced ad libitum energy intake independent of changes in peripheral plasma PYY and GLP-1. Thus, the underlying mechanism for IPE’s appetite-suppressing effect remains unknown.

## Figures and Tables

**Figure 1 nutrients-11-00861-f001:**
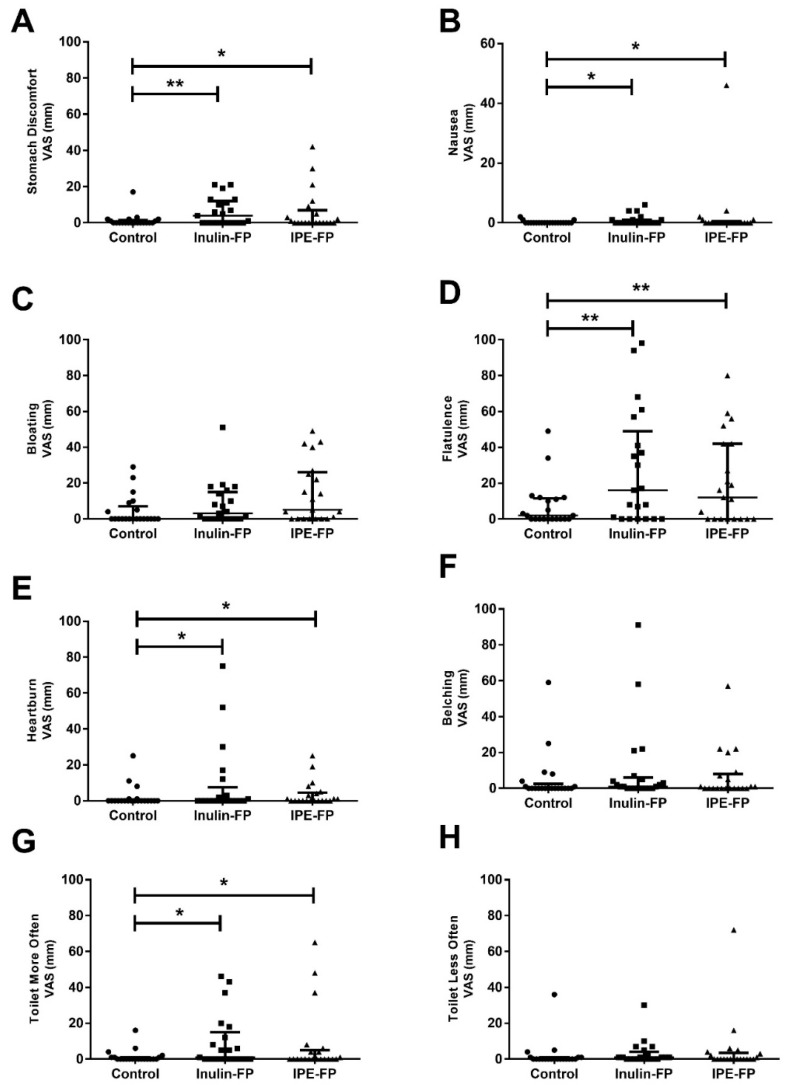
Gastrointestinal side effects following a seven-day supplementation with inulin-FP or IPE-FP compared to control-FP. Data represents median [IQR] gastrointestinal side-effects ratings (**A**) Stomach Discomfort, (**B**) Nausea, (**C**) Bloating, (**D**) Flatulence, (**E**) Heartburn, (**F**) Belching, (**G**) Toilet More Often and (**H**) Toilet Less Often, which were made using 100 mm VAS (*n* = 21). Subjects were asked to rate the occurrence of each side effect with extreme statements anchored at each end of the rating scale (0 mm Never, 100 mm All the time). * and ** indicate a significance between treatments of *p* < 0.05 and *p* < 0.01, respectively, in post-hoc tests. Abbreviations: VAS, visual analogue scale.

**Table 1 nutrients-11-00861-t001:** Baseline characteristics of subjects and changes in anthropometric measurements and blood lipid profiles between study visits.

	Control-FP ^4^	Inulin-FP ^5^	IPE-FP ^6^	P ^7^
Male n (%)	9/21 (43%)			
Age (year)	60 [53, 64]			
Weight (kg)	80.2 [76.1, 88.4]	80.5 [75.9, 88.5]	81.7 [76.2, 89.5] ^# $^	**0.023**
FFM (kg)	51.5 [46.2, 66.0]	50.9 [46.5, 67.2]	51.6 [47.3, 67.5] ^##^	**0.024**
FM (kg)	29.1 [23.3, 33.7]	28.8 [23.3, 32.8]	29.2 [24.2, 32.8]	0.48
% FFM	65 [59.9, 74.0]	65.5 [60.3, 74.9]	64.8 [60.1, 74.3]	0.16
% FM	35 [26.0, 40.2]	34.5 [25.2, 40.0]	35.2 [25.8, 40.0]	0.15
TBW	37.7 [33.9. 48.3]	37.2 [34.1, 49.2]	37.8 [34.6, 49.4] ^## $^	**0.026**
Fasting cholesterol (mmol/L)				
Total	5.1 [4.8, 6.4]	5.0 [4.6, 6.1]	5.3 [4.7, 5.9]	0.86
HDL	1.5 [1.3, 1.9]	1.4 [1.2, 1.7]	1.5 [1.3, 1.8]	0.92
LDL	3.1 [2.5, 4.1]	3.0 [2.3, 4.0]	3.1 [2.5, 3.8]	0.95
Total Chol:HDL	3.6 [2.8, 4.6]	3.7 [2.9, 4.6]	3.8 [2.9, 4.4]	0.97
LDL:HDL	2.1 [1.6, 3.1]	2.3 [1.6, 2.9]	2.4 [1.6, 2.7]	0.87
Fasting TG (mmol/L)	1.2 [1.0, 1.6]	1.2 [1.0, 1.7]	1.3 [0.9, 1.7]	0.10
Power of food scale	35 [26, 42]	33 [25, 44]	33 [24, 46]	0.68
IPAQ ^1,2^	4230 [2025, 6115]	3570 [2280, 6678]	3750 [1504, 5190]	0.85
24 h food intake (kcal) ^3^	2250 [1681, 2737]	1948.5 [1800, 2358]	2222 [1775, 2547]	0.95

Data are presented as median [IQR]. Data represent *n* = 21, except for ^1^ where *n* = 14. ^2^ IPAQ captured data relating to daily activity from the previous week. ^3^ Subjects completed diet diaries on the day before each study visit, which were assessed using Dietplan 6 software. ^4^ Control data were collected on the control study visit where participants consumed unsupplemented food products. ^5^ Inulin-FP and ^6^ IPE-FP data were collected on the post-supplementation visits. ^7^ Data compared using a Friedman test with significant differences highlighted in bold. ^#^ and ^##^ indicate a significance between Control-FP and IPE-FP of *p* < 0.05 and *p* < 0.01, respectively, in post-hoc tests. ^$^ indicates a significance between Inulin-FP and IPE-FP of *p* < 0.05 in post-hoc tests. Abbreviations: BMI, body mass index; FFM, fat-free mass; high-density lipoproteins, HDL; IPAQ; International Physical Activity Questionnaire; IPE, inulin-propionate ester; FP, food products; LDL, low-density lipoprotein; NEFA, non-esterified fatty acids; TBW, total body water; TG, triglycerides.

**Table 2 nutrients-11-00861-t002:** Ad libitum energy intake, gut hormone and composite appetite score measures following a seven-day supplementation with inulin-FP or IPE-FP compared to control-FP.

	Unit	Value	Control-FP ^2^	Inulin-FP ^3^	IPE-FP ^4^	P ^5^
Energy intake	kcal		830.8 [643.7, 1158.3]	878.5 [415.4, 1112]	699.2 [398.6, 1053.3]	**0.012**
PYY	pmol/L	0 min	47.30 [29.35, 57.69]	44.29 [28.29, 54.91]	46.16 [29.85, 61.50]	0.41
		AUC_0-420_	53.41 [43.6, 70.31]	58.51 [42.88, 70.80]	54.86 [44.91, 68.16]	0.95
GLP-1	pmol/L	0 min	20.98 [14.72, 28.04]	18.63 [16.33, 22.50]	20.08 [13.34, 24.78]	0.26
		AUC_0-420_	22.73 [18.55, 26.34]	22.88 [17.60, 24.94]	21.77 [17.67, 26.18]	0.10
CAS ^1^	mm	0 min	50.0 [40.0, 70.5]	58.0 [41.5, 79.0]	50.0 [37.0, 76.5]	0.86
		AUC_0-420_	25.7 [11.8, 43.5]	23.3 [10.2, 45.0]	34.9 [15.8, 41.3]	0.28

Data are presented as median [IQR]. Data represent *n* = 21, except for ^1^ where *n* = 19. ^2^ Control data were collected on the control study visit where participants consumed unsupplemented food products. ^3^ Inulin-FP and ^4^ IPE-FP data were collected on the post-supplementation visits. ^5^ Data compared using a Friedman test with significant differences highlighted in bold. Abbreviations: AUC, area under the curve; CAS, composite appetite score; IPE, inulin-propionate ester; FP, food products.

**Table 3 nutrients-11-00861-t003:** Food product satisfaction ratings for inulin-FP and IPE-FP compared to control-FP ^1^.

Food Product	Control-FP ^2^	Inulin-FP ^3^	IPE-FP ^4^	P ^5^
Bread				
Overall	6 [4, 8]	6 [4, 8]	6 [4, 8]	0.70
Appearance	5 [5, 7]	6 [5, 7]	6 [5, 7]	0.46
Aroma	6 [5, 7]	6 [5, 7]	6 [5, 7]	0.85
Flavour	7 [5, 8]	6 [5, 8]	6 [4, 8]	0.29
Texture	5 [4, 7]	5 [4, 7]	5 [4, 7]	0.79
Smoothie				
Overall	7 [7, 8]	7 [6, 8]	7 [5, 8]	0.53
Appearance	6 [5, 8]	7 [5, 8]	7 [5, 8]	0.89
Aroma	6 [5, 8]	7 [5, 8]	6 [5, 7]	0.50
Flavour	7 [7, 8]	7 [6, 9]	6 [4, 8]	0.25
Texture	7 [6, 8]	7 [4, 7]	7 [6, 8]	0.52

Data are presented as median [IQR] (*n* = 21). ^1^ Participants rated the product overall, and based on its appearance, aroma, flavour and texture. Participants ticked a box between 1 and 9, where 1 was labelled with “dislike extremely”, 5 was labelled with “neither like nor dislike” and 9 was labelled with “like extremely”. ^2^ Control data were collected on the control study visit where participants consumed unsupplemented food products. ^3^ Inulin-FP and ^4^ IPE-FP data were collected on the post-supplementation visits. ^5^ Data were compared using a Friedman test. Abbreviations: FP, food products; IPE, inulin-propionate ester.

**Table 4 nutrients-11-00861-t004:** Fasting values and AUC for energy expenditure and substrate oxidation, and after correcting for fat-free mass, following a seven-day supplementation with inulin-FP or IPE-FP compared to control-FP.

	Unit	Value ^1^	Control-FP ^2^	Inulin-FP ^3^	IPE-FP ^4^	P ^5^
RER	VCO_2_/VO_2_	0 min	0.762 [0.725, 0.787]	0.764 [0.712, 0.824]	0.766 [0.683, 0.813]	0.71
		AUC_0-240_	0.823 [0.756, 0.841]	0.825 [0.773, 0.875]	0.842 [0.745, 0.859]	0.95
Raw						
REE	kcal/min	0 min	1.031 [0.936, 1.179]	1.103 [0.909, 1.197]	1.121 [0.974, 1.276]	**0.051**
		AUC_0-240_	1.144 [1.073, 1.271]	1.152 [1.056, 1.275]	1.255 [1.091, 1.321] ^# $^	**0.018**
CHO Ox.	g/min	0 min	0.037 [-0.024, 0.051]	0.041 [−0.015, 0.083]	0.042 [−0.058, 0.091]	0.87
		AUC_0-240_	0.094 [0.024, 0.136]	0.097 [0.056, 0.162]	0.102 [0.013, 0.163]	0.78
Lipid Ox.	g/min	0 min	0.071 [0.055, 0.089]	0.076 [0.038, 0.085]	0.078 [0.046, 0.115]	0.86
		AUC_0-240_	0.054 [0.033, 0.072]	0.050 [0.032, 0.082]	0.052 [0.032, 0.099]	0.33
Protein Ox.	g/min	total	0.057 [0.049, 0.065]	0.058 [0.042, 0.062]	0.057 [0.042, 0.065]	0.95
FFM corr. ^6^						
REE	kcal/min/kg FFM	0 min	0.020 [0.016, 0.022]	0.020 [0.016, 0.022]	0.021 [0.017, 0.022]	0.37
		AUC_0-240_	0.021 [0.017, 0.025]	0.021 [0.018, 0.024]	0.022 [0.018, 0.024]	0.22
CHO Ox.	mg/min/kg FFM	0 min	0.718 [−0.346, 0.940]	0.734 [-0.245, 1.751]	0.663 [−1.028, 1.364]	0.95
		AUC_0-240_	1.849 [0.388, 2.817]	1.881 [1.087, 3.073]	1.900 [0.159, 3.011]	0.83
Lipid Ox.	mg/min/kg FFM	0 min	1.470 [0.934, 1.652]	1.238 [0.789, 1.634]	1.345 [0.950, 1.770]	0.87
		AUC_0-240_	1.024 [0.638, 1.272]	1.006 [0.614, 1.365]	0.921 [0.640, 1.610]	0.72
Protein Ox.	mg/min/kg FFM	total	1.080 [0.869, 1.261]	0.965 [0.817, 1.216]	0.978 [0.810, 1.199]	0.67

Data are presented as median [IQR)] (*n* = 21). ^1^ AUC were calculated using the trapezoidal rule and then divided by the relevant time period (240 min). ^2^ Control data were collected on the control study visit where participants consumed unsupplemented food products. ^3^ Inulin-FP and ^4^ IPE-FP data were collected on the post-supplementation visits. ^5^ Data represents the difference between post-supplementation and control measurements. ^5^ Data compared using a Friedman test with significant differences and trends highlighted in bold. ^6^ REE and substrate oxidation measurements after correcting for FFM. ^#^ indicates a significance between Control-FP and IPE-FP and ^$^ indicates a significance between Inulin-FP and IPE-FP of *p* < 0.05 in post-hoc tests. Abbreviations: AUC, area under the curve; CHO, carbohydrate; corr., corrected; FFM, fat free mass; FP, food products; IPE, inulin-propionate ester; Ox, oxidation; REE, resting energy expenditure; RER, respiratory exchange ratio.

**Table 5 nutrients-11-00861-t005:** Fasting values and AUC for breath H_2_, blood hormones and metabolites following a seven-day supplementation with inulin-FP or IPE-FP compared to control-FP.

	Unit	Value ^1^	Control-FP ^2^	Inulin-FP ^3^	IPE-FP ^4^	P ^5^
Breath H_2_	ppm	0 min	4.0 [1.5, 11.0]	5.0 [2.0, 11.0]	3.0 [1.5, 11.0]	0.44
		AUC_0-420_	11.0 [6.0, 20.5]	18.4 [7.5, 31.6] **	11.7 [6.2, 20.0] ^$^	**0.010**
		AUC_0-240_	6.1 [3.2, 9.3]	7.4 [5.2, 14.8]	7.4 [4.6, 13.1]	0.54
		AUC_240–420_	17.8 [7.9, 27.8]	32.3 [12.0, 50.6] **	16.7 [7.7, 31.6] ^$^	**0.004**
Glucose	mmol/L	0 min	5.43 [5.06, 5.74]	5.44 [5.25, 5.76]	5.33 [5.01, 5.70]	0.87
		AUC_0-420_	6.00 [5.61, 6.91]	6.13 [5.63, 6.60]	6.10 [5.88, 6.81]	0.17
Insulin	µU/ml	0 min	2.28 [2.04, 2.70]	2.34 [2.26, 2.71]	2.37 [2.16, 2.61]	0.10
		AUC_0-420_	48.20 [37.44, 57.51]	51.8 [40.15, 61.10]	54.49 [39.10, 59.75]	0.41
NEFA	mmol/L	0 min	0.71 [0.59, 0.83]	0.66 [0.48, 0.77]	0.64 [0.40, 0.74]	0.11
		AUC_0-420_	0.14 [0.10, 0.18]	0.11 [0.08, 0.16]	0.11 [0.08, 0.15]	0.16
Acetate	µmol/L	0 min	51.75 [41.27, 61.05]	54.46 [42.50, 74.29]	54.90 [34.90, 73.19]	0.95
		AUC_0-420_	58.72 [36.64, 73.71]	56.35 [45.23, 72.60]	57.64 [44.18, 76.70]	0.18
Propionate	µmol/L	0 min	2.20 [1.53, 2.54]	2.37 [1.88, 2.88]	2.30 [1.70, 3.06]	0.26
		AUC_0-420_	1.92 [1.68, 2.78]	2.27 [1.63, 2.53]	2.38 [2.04, 2.83]	0.10
Butyrate	µmol/L	0 min	1.17 [0.46, 1.45]	1.29 [0.94, 1.72]	1.04 [0.67, 1.64]	0.41
		AUC_0-420_	1.21 [0.83, 1.56]	1.11 [0.96, 1.36]	1.28 [1.00, 1.62]	0.47

Data are presented as median [IQR] (*n* = 21). ^1^ AUC were calculated using the trapezoidal rule and then divided by the relevant time period (180 or 240 min) with AUC_0-420_ representing the total AUC and the AUC_0-240_ and AUC_240–420_ representing before and after >80% IPE has been shown to enter the colon, respectively [10]. ^2^ Control data were collected on the control study visit where participants consumed unsupplemented food products. ^3^ Inulin-FP and ^4^ IPE-FP data were collected on the post-supplementation visits. ^5^ Data compared using a Friedman test with significant differences highlighted in bold. ** indicates a significance between Control-FP and Inulin-FP of *p* < 0.01 in post-hoc tests. ^$^ indicates a significance between Inulin-FP and IPE-FP of *p* < 0.05 in post-hoc tests. Acetate, propionate and butyrate were measured according to a previously published method [20]. Abbreviations: AUC, area under the curve; CAS, composite appetite score; FP, food products; GLP-1, glucagon like peptide-1; IPE, inulin-propionate ester; NEFA, non-esterified fatty acids; PYY, peptide YY.

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
