# Peer review of "Effects of Inulin Propionate Ester Incorporated into Palatable Food Products on Appetite and Resting Energy Expenditure: A Randomised Crossover Study"

_nutrients, 2019, doi:10.3390/nu11040861_

Round 1
Reviewer 1 Report
This is an interesting study of a functional ingredient, IPE, that builds the body of evidence for enhanced fibers. The study design is adequate and appropriate. The authors need to make minor revisions to the paper.
Throughout the figures and tables: Per the narrative, numbers in the same row with the SAME letter notations are significantly DIFFERENT. Prefer that significant differences are communicated with DIFFERENT letters. Regardless of the convention, the meaning of the letters needs to be more clearly described in the foot note of all tables and figures with significant differences. The current foot notes are not descriptive enough.
Data communication with IQR makes sense for some values, but I struggle to see the need to report IQR instead of SD or SEM for anthropometric and standard biochemical data. IQR makes it difficult to compare the variation reported in this study with other previously published studies. Understandably, IQR may be appropriate for the nonparametric outcomes, but I struggle to believe that all of the outcomes were nonparametric, and if so, this creates an inherent issue in the sample.
Line 29 Abstract. The statement “In order to be used as an effective public health tool, IPE…” implies that this ingredient should be widely supplemented to the population. The manuscript does not provide enough evidence in the present study, nor does it summarize a compelling case based on previous literature, that this is a functional ingredient that all ages, all demographics should be consuming. Please revise.
Per line 440, the propionate is expected to be delivered to the colon because it is esterified to the inulin. Would this create different physiological effects/different mechanism due to colonic delivery versus small intestine delivery? This is important to reflect in the discussion of previous studies on weight gain and REE in line 410 and following.
Supplementary table 1
Include the portion size for each food
Are the foods described in table 1 the unsupplemented foods? Please clarify in the table. The composition of the supplemented foods is more relevant.
Supplementary figure 1
Add the length of wash out to the figure or as a footnote.
A table 5 should be moved to the beginning of the manuscript because it provides critical data for understanding the characteristics of the subjects. Regarding body weight, the differences in BW and FFM need to be contextualized with water intake and retention. The significant differences may not actually represent physiologically relevant changes in body weight
The GI symptoms are a significant finding and should be moved from the supplementary materials to the main manuscript. Timing of VAS for gastrointestinal symptoms needs to be described. Its unclear if this was measured with the appetite VAS. The data presented in Supp Fig 2 suggest that it was a one-time rating.
The subject satisfaction ratings for the food products are less critical, so this could be put in the supplementary materials to provide room for the GI symptoms in the main manuscript. Why was a Likert-type rating used for the subject satisfaction questionnaire rather than a VAS?
Line 125- The added rolls and smoothies provided 1621 kJ energy per day. Were the subjects counseled on how to replace existing diet items with the study foods? If not, the limitations of not counseling the subjects needs to be discussed later in the paper. Further details on dietary intake, beyond just energy, need to be reported. This could be in the supplementary materials.
Author Response
Please find our response to the reviewer attached.

Reviewer 2 Report
Comments to Nutrients manuscript 465964 “Effects of inulin propionate ester incorporated into 2 palatable food products on appetite and resting 3 energy expenditure: a randomised crossover study”
Overall assesment
The use of propionate could be an interesting tool to reduce food intake, so the topic is interesting. However I have some doubt about the reliability of data. The treatment did not changed propionate AUC and it is quite strange, even though liver can sequestrate propionate for gluconeogenesis. But glucose and insulin resulted not affected by treatment.
In addition the way to present data in some tables must be improved.
In my opinion the paper need MAYOR REVISION.
Mayor remarks
Why is the body weight increased in the IPE group, if the energy intake was reduced following the supplementation with IPE ?
The propionate levels were not different between treatments, how is possible ? Since the AUC of propionate between the treatments were not different, the explanation reported at lines 400-401 is not correct.
Minor remarks
Lines 33 and 80 In the abstract the experimental period lasted 6 days, while in the Introduction (line 80) the experimental period is said to be 7 days. Which is the right duration ?
Table 3 The model for differences in REE expressed as AUC 0-240 was significant but the means reported as significantly differ were only Control and Inulin, which have similar values, while IPE, that determined an higher REE, resulted not different compared to the other two treatments.
Table 4 According to this table, the AUC for Breath H2 has not a significant trend for 0-240, but Authors reported significant differences between Inulin-FP and IPE-FP even though the means are very similar. If P-value of model is not significant, no differences between treatments should be reported.
Table 5 As for table 4, if the P-value for body weight is significant, the different superscript should be placed at higher and the lower mean, but this has not been done.
Author Response
Please see our response to the reviewer attached.

Round 2
Reviewer 2 Report
OK, Authors have modified the manuscript, which is now acceptable for the publication.
Author Response
Many thanks for your helpful comments.